# EEG Fingerprints under Naturalistic Viewing Using a Portable Device

**DOI:** 10.3390/s20226565

**Published:** 2020-11-17

**Authors:** Matteo Fraschini, Miro Meli, Matteo Demuru, Luca Didaci, Luigi Barberini

**Affiliations:** 1Department of Electrical and Electronic Engineering, University of Cagliari, 09123 Cagliari, Italy; m.meli4@studenti.unica.it (M.M.); ldidaci@unica.it (L.D.); 2Stichting Epilepsie Instellingen Nederland (SEIN), 2103SW Heemstede, The Netherlands; matteo.demuru@univ-amu.fr; 3Department of Medical Sciences and Public Health, University of Cagliari, 09123 Cagliari, Italy; barberini@unica.it

**Keywords:** EEG, fingerprints, emotion, spectral analysis, naturalistic stimuli

## Abstract

The electroencephalogram (EEG) has been proven to be a promising technique for personal identification and verification. Recently, the aperiodic component of the power spectrum was shown to outperform other commonly used EEG features. Beyond that, EEG characteristics may capture relevant features related to emotional states. In this work, we aim to understand if the aperiodic component of the power spectrum, as shown for resting-state experimental paradigms, is able to capture EEG-based subject-specific features in a naturalistic stimuli scenario. In order to answer this question, we performed an analysis using two freely available datasets containing EEG recordings from participants during viewing of film clips that aim to trigger different emotional states. Our study confirms that the aperiodic components of the power spectrum, as evaluated in terms of offset and exponent parameters, are able to detect subject-specific features extracted from the scalp EEG. In particular, our results show that the performance of the system was significantly higher for the film clip scenario if compared with resting-state, thus suggesting that under naturalistic stimuli it is even easier to identify a subject. As a consequence, we suggest a paradigm shift, from task-based or resting-state to naturalistic stimuli, when assessing the performance of EEG-based biometric systems.

## 1. Introduction

During the last several years, the electroencephalogram (EEG) has been proven to be a promising technique for personal identification and/or verification [1]. Despite its noisy characteristics, several scalp EEG features still contain relevant subject-specific traits that have been tested under any conceivable scenario. In particular, EEG fingerprints have been successfully observed and reported in resting-state [2,3,4], motor, visual, auditory, imagined speech or even multi-functional systems [5,6,7]. More recently, the aperiodic component of the power spectrum [8], as defined by the offset and the exponent, which reflect its 1/f-like characteristic, was shown to outperform other commonly used EEG features [9].

Beyond that, EEG characteristics may be sensitive to or may capture relevant features related to emotional states [10,11,12,13], which may also play an important role in the development of the brain-computer interfaces (BCI) [14]. Although there is ample literature about this last topic, the possible effects of emotional states in EEG-based identification systems have not been widely investigated. Nevertheless, some recent findings suggest that the influence of emotional states on EEG biometric systems should be properly taken into account [15,16].

In this work, we aim to understand if the aperiodic component of the power spectrum, used in the resting-state experimental paradigm [9], is able to capture EEG-based subject-specific features in naturalistic stimuli scenarios. In order to answer this question, we performed an analysis using a freely available EEG dataset [17] containing EEG recordings from 23 participants during the viewing of film clips that aim to trigger different emotional states. Moreover, we have replicated our results using another EEG dataset from 32 participants also recorded during naturalistic viewing (i.e., movie watching) [18]. Such naturalistic paradigms, which represent a better approximation of real-life scenarios, using stimuli such as films, spoken narratives, or music emerged in response to the common concerns about the use of simple tasks or no-tasks resting-state paradigms [19]. The analysis was performed using the FOOOF tool [8] that works on the frequency representations (power spectra), fits the model, shows original power spectrum with the associated model fit, and provides the parameters of the model fit, namely the aperiodic components (offset and exponent). Moreover, the study was also replicated using the classical analysis performed in terms of the more common periodic components, namely theta (4–8 Hz), alpha (8–13 Hz), beta (13–30 Hz), and gamma (30–45 Hz) frequency bands. For both approaches, the performance evaluation procedure was performed using a standard approach to evaluate biometric systems [20].

We would like to highlight that the aim of this study is twofold. It represents a verification of the stability and robustness of the aperiodic component of the power spectrum in picking up relevant subject-specific features over several and different naturalistic stimuli, which, to the best of our knowledge, was never evaluated before and, moreover, we argue that this is also important to realize how individual variability may be relevant when automated emotion recognition represents the ultimate goal.

## 2. Materials and Methods

### 2.1. Datasets

In this work, we used DREAMER [17], a freely available dataset for emotion recognition from wireless low-cost devices, which includes EEG signals from 23 participants recorded during affect elicitation by means of audio-visual stimuli. The EEG signals were recorded using a sampling rate of 128 Hz with an Emotiv EPOC fourteen-channel system. Emotions were elicited by using 18 film clips that have been shown to evoke a wide range of emotions such as amusement, excitement, happiness, calmness, anger, disgust, fear, sadness, and surprise. Together with the EEG signals, the participants’ self-assessment of their affective state after each stimulus (in terms of valence, arousal, and dominance) was also acquired. Our analysis was performed on two different segments, each one lasting 1 min. The first segment (video_start) includes the first 60 s of the stimulus and the second segment (video_end), which represents a replication of the study, includes the last 60 s of the same stimulus. The reason for reproducing the analysis using two different segments was to understand how the film clip length may play a significant role in defining the final results [19]. Moreover, as for the video stimuli, the analysis was further replicated using a baseline condition, which represents a 60 s eyes-open resting-state condition, where a neutral clip was shown in order to help the subject return to a neutral emotional state. The whole analysis was later replicated using another EEG dataset, the DEAP dataset, a dataset for emotion analysis using EEG, physiological, and video signals [18], where the EEG and peripheral physiological signals of 32 participants were recorded as each watched 40 one-minute-long music videos. The data also contain, for each single participant, a baseline recording (eyes-open resting-state) that we used to compare the differences between the two conditions (i.e., resting-state and naturalistic viewing). For both analyses, each segment was successively organized into non-overlapping epochs of 15 s [21].

### 2.2. Features Extraction

For each subject, film clip, segment, and epoch, we computed the aperiodic components of the power spectrum, reflecting 1/f-like characteristics, namely the offset and the exponent, using the FOOOF tool [8]. In particular, the FOOOF algorithm works on frequency representations (power spectra in linear space), fits the model, shows the original power spectrum with the associated model fit, and provides the parameters of the model fit, the aperiodic components (offset and exponent). The FOOOF tool is freely available both for MATLAB and Python [8]. Before computing the aperiodic components, the raw EEG signals were filtered using a band-pass filter between 1 and 50 Hz using the ‘eegfilt’ function in EEGLAB [22] and subsequently the power spectral density was estimated using the ‘pwelch’ method in MATLAB (The MathWorks, Inc., Natick, MA, USA, version 9.8.0.1323502, R2020a). Finally, for each film clip and each subject, we obtained a feature vector, consisting of fourteen values (one value for each EEG channel), separately for the two aperiodic components. The relevance of these parameters in the context of this study and the implementation of the aperiodic components as EEG fingerprints in a different experimental scenario (resting-state paradigm) were previously investigated and reported in [9]. Finally, we compared our results with the classical analysis performed using the common periodic components, namely theta (4–8 Hz), alpha (8–13 Hz), beta (13–30 Hz) and gamma (30–45 Hz) frequency bands.

### 2.3. Performance Evaluation and Statistical Analysis

The performance of the two aperiodic components was obtained using a standard approach used to evaluate biometric systems [20]. The approach requires the definition of genuine and impostor scores (equal to 1/(1 + d), where d is the Euclidean distance), computed between pairs of feature vectors. In more detail, for each feature vector, consisting of fourteen values (one value for each EEG channel), we computed the similarity score against every other feature vector, thus obtaining the genuine scores (within-subject) and impostor score (between-subjects) distributions. The overall performance was successively evaluated from the false acceptance rate (FAR, the error occurring when an impostor is accepted) and the false rejection rate (FRR, the error occurring when a genuine is rejected) using different thresholds. The equal error rate (EER), the point where FAR equals FRR, was reported to outline the final results, so that low EER values represent high performance. The Wilcoxon rank test was used in order to test possible statistical differences among the different experimental scenarios. The statistical results are reported in terms of *p*-value, confidence interval, and effect size. All the analysis was performed using MATLAB (The MathWorks, Inc., Natick, MA, USA, version 9.8.0.1323502, R2020a) and all the figures were realized using Jamovi (version 1.0.8.0) available from https://www.jamovi.org. All the scripts used to perform the analysis are freely available at the following link: https://github.com/matteogithub/EEG-fingerprints-for-Sensors.

## 3. Results

### 3.1. 1/f Offset Parameter

A visual representation of the extracted feature vectors for a single video-clip are represented in Figure 1. The best absolute performance, in terms of EER, was observed during naturalistic stimuli as shown in Figure 2. In particular, for the video_start condition, we obtained an EER equal to 0.08 for an excitement target video, which represents overall the best performance found during this analysis. The results from corresponding statistical analysis, including *p*-values, confidence intervals, and effect sizes, performed using a Wilcoxon rank test, are summarized in Table 1. As reported, we observed significant differences between baseline and task conditions, irrespective of the time window (first or last part of the experiment) used for the analysis.

### 3.2. 1/f Exponent Parameter

Again, the best absolute performance, in terms of EER, was observed during naturalistic stimuli as shown in Figure 3. As for the 1/f offset parameter, for the video_start condition, we obtained an EER equal to 0.12 for an amusement target video, which, however, represents a lower performance if compared with the 1/f offset parameter. The results from corresponding statistical analyses, including *p*-values, confidence intervals, and effect sizes, performed using a Wilcoxon rank test, are summarized in Table 2. As reported, we observed significant differences between baseline and task conditions, irrespective of the time window (first or last part of the experiment) used for the analysis. Furthermore, a significant difference was also observed between the first part and the last part of the baseline.

### 3.3. Correlation with Participants’ Self-Assessment

Finally, we also observed moderate correlations between the dominance (computed as the mean over all the subjects) and the EER for the offset parameter, both for the first (rho = −0.477, *p* = 0.045) and the second (rho = 0.412, ns) block, using the non-parametric Spearman method.

### 3.4. Periodic Components

As for the periodic components, we observed the following performance in terms of EER for the first 60 s of film clip viewing: 0.24 ± 0.03 for the theta band, 0.24 ± 0.04 for the alpha band, 0.15 ± 0.02 for the beta band, and 0.19 ± 0.03 for the gamma band. Moreover, we observed the following performance in terms of EER for the last 60 s of film clip viewing: 0.25 ± 0.04 for the theta band, 0.23 ± 0.04 for the alpha band, 0.14 ± 0.03 for the beta band, and 0.17 ± 0.03 for the gamma band. Finally, we observed the following performance in terms of EER for the baseline condition: 0.28 ± 0.02 for the theta band, 0.28 ± 0.02 for the alpha band, 0.20 ± 0.03 for the beta band, and 0.24 ± 0.03 for the gamma band. The differences among the three conditions, namely baseline, video_start and video_end, for the beta band, which was the periodic component with the higher performance, are represented in Figure 4. The corresponding statistics are reported in Table 3.

### 3.5. Replication on the DEAP Dataset

In the replication part of the study, we observed the best absolute performance, in terms of EER, during naturalistic stimuli using the offset parameter, with a minimum EER value equal to 0.099 and a mean of 0.166 ± 0.031. The exponent parameter performed worse, with a minimum EER value equal to 0.201 and a mean of 0.285 ± 0.030. Therefore, as for the original results, the use of video clips outperformed the use of resting-state paradigms, where, for this latter approach the EER was equal to 0.350 for the offset and equal to 0.362 for the exponent. It is important to highlight that, for this second dataset we have only one resting-state (i.e., baseline) recording for each subject since these traces were recorded before the video clips were presented. As for the periodic component of the power spectra, again, the naturalistic stimuli outperform the baseline condition for 3 out of 4 frequency bands, namely theta (minimum EER value equal to 0.147 and a mean of 0.184 ± 0.026 for the naturalistic stimuli and EER value equal to 0.194 for the baseline), alpha (minimum EER value equal to 0.159 and a mean of 0.204 ± 0.023 for the naturalistic stimuli and EER value equal to 0.233 for the baseline), and beta (minimum EER value equal to 0.146 and a mean of 0.194 ± 0.029 for the naturalistic stimuli and EER value equal to 0.147 for the baseline), where the opposite was observed for the gamma band (minimum EER value equal to 0.249 and a mean of 0.285 ± 0.022 for the naturalistic stimuli and EER value equal to 0.173 for the baseline).

## 4. Discussion

In summary, this study confirms that the aperiodic components of the power spectrum [8,9], as evaluated in terms of offset and exponent parameters, are able to detect subject-specific features extracted from the scalp EEG.

In particular, our results show that the performance of the system was significantly higher (lower EER value) for the film clip scenario, thus suggesting that under a naturalistic stimulus it is even easier to identify a subject. This is of special relevance since naturalistic stimuli represent a better approximation of real-life scenarios [19] if compared with the more classical approaches based on arbitrary tasks or resting-state paradigms. Furthermore, our results also show a moderate correlation between the system performance and the participants’ self-assessment, especially for the dominance scale where the magnitude of the association was particularly high and significant, at least for the offset parameter.

Moreover, we reproduced the study using the more common approach based on the classical decomposition of the EEG signals in frequency contents, namely theta, alpha, beta, and gamma bands. In this case, it is of relevance to highlight that, as expected [3,9], the high frequency contents, beta and gamma bands, represent the periodic components that gave the best performance, in all the investigated scenarios. Furthermore, we argue that this approach represents an important replication of the results obtained using the aperiodic components, since, even in this case, we successfully demonstrated that the naturalistic stimuli, as provided using film clips, outperform the use of no-task conditions. Finally, it is also interesting to note that, as previously shown [9], the aperiodic component of the power spectrum outperforms the use of the more classical approach based on periodic components (frequency bands decomposition).

In order to understand to what extent the reported results are dependent on the specific set of data used in this study, we decided to replicate the whole analysis using another fresh dataset. Specifically, we used the DEAP dataset [18], a dataset for emotion analysis using EEG, physiological, and video signals, which have several similarities with the original one. Interestingly, we may conclude that we successfully replicated the original findings and thus we are more confident that the reported results can be considered robust, and at least in part generalizable.

We also want to emphasize that a recent work [23], even though performed using the functional magnetic resonance technique, shows that compared to the resting-state, naturalistic viewing allows a more accurate prediction of trait-like phenotypes in both cognitive and emotional domains, also suggesting that naturalistic stimuli amplify individual differences. In line with this result, our study, even using a different recording technique and a different set of features, confirms that naturalistic viewing allows the increase of the performance in a subject identification scenario such as the one explored in this work. In this context, our study suggests that even with a much lower spatial resolution, as represented by the use of low-density scalp-EEG, the naturalistic stimuli, as observed in functional magnetic resonance, amplify individual differences.

Nevertheless, we recognize two important limitations of the present study. The first limitation concerns the number of subjects involved in the study since we used a freely available dataset, which on the other hand represents a strong point in terms of reproducibility, and it is not straightforward to collect such complex data; this limitation is not easy to overcome. However, we think that future studies should aim to replicate these results on a bigger dataset in order to generalize these findings. The second limitation concerns the use of scalp EEG recordings, moreover from a low-cost device, which is very well known to be affected by diverse sources of noise. We argue that, even if not relevant for biometric systems, it would be important to investigate whether these findings still hold when using source-reconstructed analysis with high-density EEG, since individual variability still represents a very important issue or limit when comparing groups or conditions [24].

The stronger point, which to the best of our knowledge was never discussed before, is that the use of naturalistic stimuli, representing experiences from everyday life and recently proposed as potential alternative to resting-state [19], outperforms the same EEG features when used in a baseline (no-stimulus) condition. We think that this result may represent valuable information for the EEG community working on individual variability and that in the near future it may drive a paradigm shift to test the performance of EEG-based biometric systems.

## 5. Conclusions

In conclusion, our study confirms, performing two completely separate analyses, that the aperiodic components of the power spectrum are able to grab subject-specific features extracted from the scalp EEG and that the performance of this approach based on naturalistic stimuli outperforms the same EEG features when used in a baseline (no-stimulus) condition. As a consequence, our results suggest focusing on naturalistic stimuli when assessing the performance of EEG-based biometric systems as this approach seems more prone to unveiling subject-specific features compared to task-based or resting-state paradigms.

## Figures and Tables

**Figure 1 sensors-20-06565-f001:**
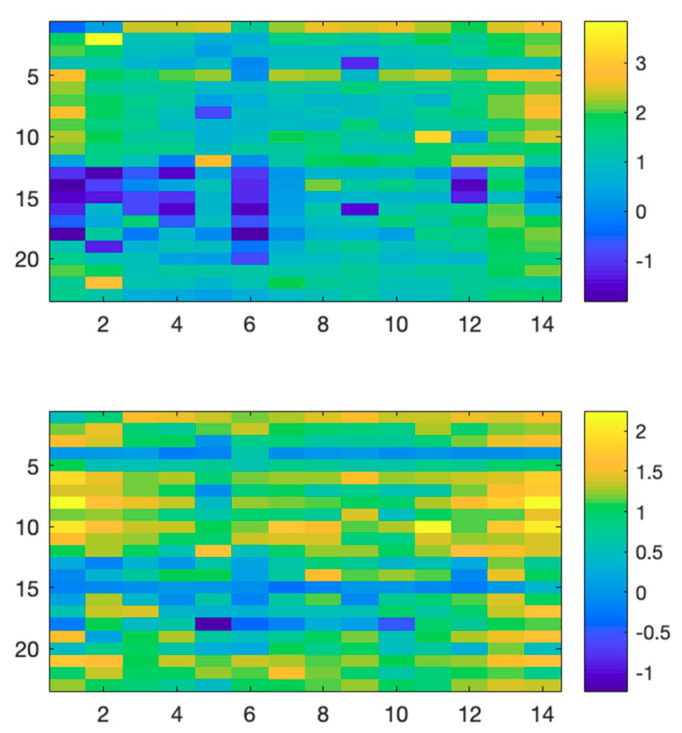
Features vector for offset (upper panel) and exponent (lower panel) computed for a single video clip extracted from the DREAMER dataset. Rows represent subjects and columns represent electroencephalogram (EEG) channels.

**Figure 2 sensors-20-06565-f002:**
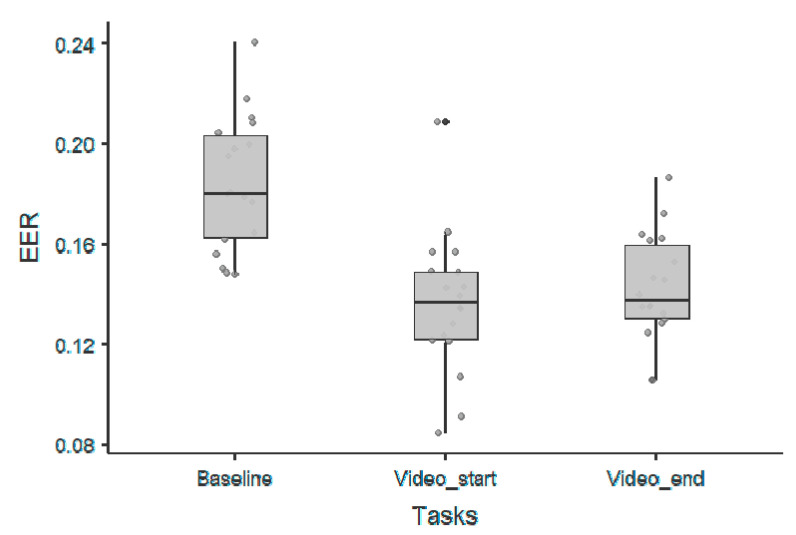
Scatter-plots represent the equal error rates (EERs) for each experimental condition for the offset parameter.

**Figure 3 sensors-20-06565-f003:**
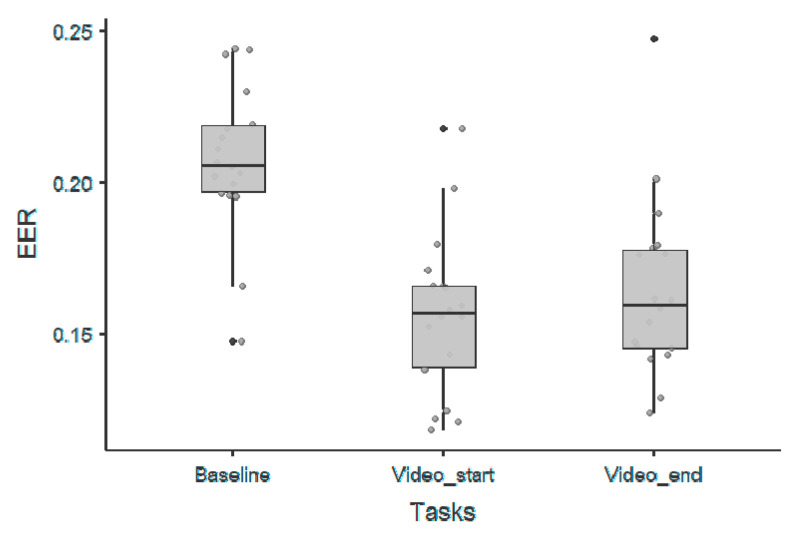
Scatter-plots represent the EERs for each experimental condition for the exponent parameter.

**Figure 4 sensors-20-06565-f004:**
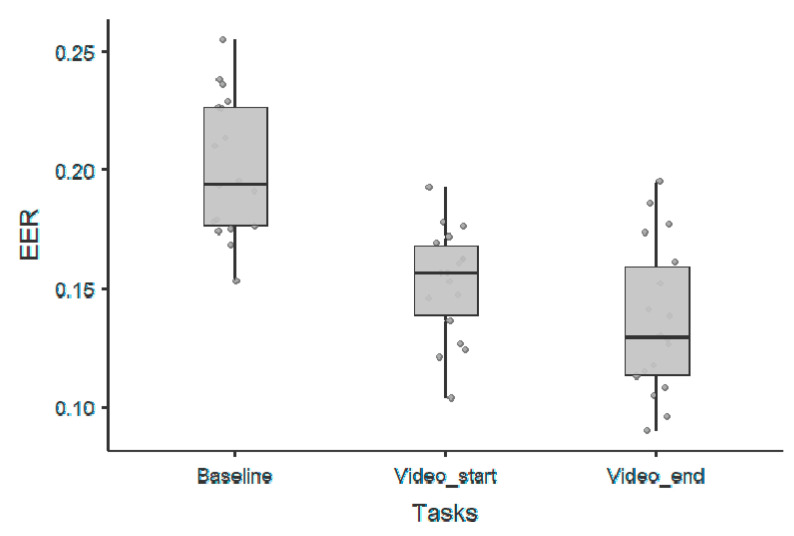
Scatter-plots represent the EERs for each experimental condition for the beta frequency band.

**Table 1 sensors-20-06565-t001:** Statistical analysis for the 1/f offset parameter.

1/f Offset Parameter—Wilcoxon Rank Test							
					95% Confidence Interval		
			Statistic	*p*	Lower	Upper	Cohen’s d
Baseline	Videos_start	Wilcoxon W	171	<0.00001	0.0345	0.05941	2.022
Baseline	Videos_end	Wilcoxon W	165	0.00011	0.0267	0.06013	1.349
Videos_start	Videos_end	Wilcoxon W	76	0.70188	−0.0222	0.00955	−0.192

**Table 2 sensors-20-06565-t002:** Statistical analysis for the 1/f exponent parameter.

1/f Exponent Parameter—Wilcoxon Rank Test							
					95% Confidence Interval		
			Statistic	*p*	Lower	Upper	Cohen’s d
Baseline	Videos_start	Wilcoxon W	171	<0.00001	0.0375	0.0636	2.174
Baseline	Videos_end	Wilcoxon W	166	0.00008	0.027	0.0602	1.371
Videos_start	Videos_end	Wilcoxon W	70	0.51354	−0.0240	0.0101	−0.250

**Table 3 sensors-20-06565-t003:** Statistical analysis for the beta band.

Beta Band—Wilcoxon Rank Test							
					95% Confidence Interval		
			Statistic	*p*	Lower	Upper	Cohen’s d
Baseline	Video_start	Wilcoxon W	168	0.00004	0.03658	0.0617	1.73
Baseline	Video_end	Wilcoxon W	169	0.00002	0.04634	0.0839	1.85
Video_start	Video_end	Wilcoxon W	129	0.05994	−0.00142	0.0357	0.479

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
