# Peer review of "EEG Fingerprints under Naturalistic Viewing Using a Portable Device"

_sensors, 2020, doi:10.3390/s20226565_

Round 1
Reviewer 1 Report
Major Comments:
In this work, the authors show how aperiodic components of the EEG power spectrum (slope and offset) can be used as a biometric identification technique. The study utilizes previously published datasets and an increasingly popular tool (FOOOF) for parameterizing the aperiodic components of the power spectrum. The evaluation techniques are standard for the field and the results demonstrate that this approach is better than using standard spectral components (at least in isolation). In general, the results are not surprising but may be of interest to the EEG-biometric community.
I have two major concerns with the manuscript in its present form. The first relates to clarity. The authors are efficient in the structure of the manuscript, but at the cost of some methodological detail and clarity. In particular, the Datasets section (2.1) should be more consistent in terminology regarding “baseline” and “resting-state”. Please clarify the baseline component (nature and duration) of each dataset (DREAMER and DEAP). Also, please elaborate on the statistical analysis approach, specifically the score generation component. Elaborate on how the pairwise comparison of the 14-dimensional feature vectors was accomplished.
My second concern has to do with generalizability. The authors correctly compare baseline and task (i.e. video clips) EEG epochs – demonstrating the lowest equal error rate (EERs) under the task/video condition. However, as far as I could tell, EEG data was from a single session for each subject. There can be significant changes in power spectral density across session, as the impedance profile of individuals change over time and are very sensitive to the exact placement of electrodes. To make a claim of generalizability, it would be imperative to calculate EERs across sessions. Is the within-subject variance of the feature vectors (across recording sessions) less that cross-subject variance?
Given the limitations of the existing datasets, this may not be possible. However, an analysis of frequency bands most susceptible to noise (beta and gamma) may shed light on the issue. The authors could compare PSD in the gamma band during the baseline and video conditions. Were there more variance-inducing artifacts during the baseline period? Likewise, a comparison alpha PSD between baseline and video conditions may indicate the degree to which unique eye movement patterns contributed to lower EER. As the authors point out, low-cost EEG systems are affected by diverse sources of noise.
Minor Comments / Questions:
Section 3.3: the correlation between EER and dominance (P = .045) is weak and may be an artifact of multiple comparisons. Did the authors compare other subjective factors and only report dominance? If so, how many and what were the rho values?
Line 200: is it demonstrably true that aperiodic outperforms periodic, for all bands? Seems like the beta band is not significantly worse the aperiodic offset/slope.
While the manuscript is succinct, there are a number of grammatical errors (e.g. run-on sentences) and unusual phrasing. A few examples are listed below, but I would encourage the authors to carefully review the manuscript.
Line 17: “film clips viewing” should be “viewing of film clips”
Line 27: “During the last years” should be “During the last several years”
Line 35: “sensible” should be “sensitive”
Line 41: “as for” should be “used in”
Line 95: “successively” should be “subsequently”
Section 3.1 and 3.2 – the “1” in 1/f has been mistakenly incorporated into the section number
Several places throughout the manuscript where “equals” should be “equal”
Line 171: “before the movie watching part” should be “before the video clips were presented”
Line 198: “proved” should be “demonstrated”
Line 199: “interesting to emphasize” should just be “interesting”
Line 206: “similitudes” should really be “similarities”
Reviewer 2 Report
The paper looks very interesting and promising. I agree with the diversion from resting-state to naturalistic stimuli if we want use EEG as biometric system. I have similar experience from my research.
- I have minor reservations about the description of the methodology and signal analysis which is a little harder to understand for people who are not closely specialized in this area. Those, as well as myself, must gradually open and read all listed references. However, after these steps the article seems scientifically correct and everything logically fits together. This is partly related to a larger number of self-citations - [2, 3, 5, 9, 21, 24]. Perhaps some literature could be replaced by inserting important parts of the cited texts directly into manuscript. This would also increase the readability of the article for people worked in different fields. I think there is still plenty of room in this paper for additional texts. The number of pages reported was 14, the actual state is 9. I sent all my comments to the author. The number of self-citations is to consider. 6 of 24 citations are self-citations. This is partly due to the narrow focus of research and the lack of a wider range of literatures. I advised the author to select important parts of his own texts and insert them to the manuscript. In this way, it could reduce the number of self-citations and make the contribution more readable and also the number of pages reported was 14, the actual state is 9. Therefore, in the manuscript is a place for additional text.
- Resolution of the images is low. If possible, increase the resolution of all images.
- There are still a few typos in the text. Please revise them. For example:
- In the titles of the chapters (“3.11. /f …”, “3.21. /f …”, correct should by “3.1. 1/f ..” and “3.2. 1/f …”)
- Line 253 “andk-core”
- Line 22, 235 “form task-based“
- In the introduction and conclusions is mentioned: "shift from task-based to naturalistic stimulus". I would either correct this statement or try to substantiate it more.
For the future: In this type of research, it will be very important to minimize all artifacts and achieve the highest possible accuracy, so it would be good to reproduce the analysis using EEG helmet with higher signal quality as also authors actually mention.
Reviewer 3 Report
"EEG fingerprints under naturalistic viewing using a portable device" is an interesting article. The purpose was to understand if the aperiodic component of the power spectrum is able to capture EEG-based subject-specific features in naturalistic stimuli scenario. The authors report that the aperiodic components of the power spectrum, as evaluated in terms of offset and exponent parameters, are able to detect subject-specific features extracted from the scalp EEG.
There are few issues in the article that need to be addressed as discussed below.
Results:
It is suggested to add a figure that shows the characteristic vectors of each of the 23 patients with the corresponding 14 values of each channel. The above will allow to visually appreciating the inter-subject differences.
Discussion:
1.- In the Discussion section suggests a more in-depth discussion of the results and their contrast with the studies carried out on the subject.
2.- In line 208 a more in-depth analysis of the results with functional magnetic resonance is suggested.
3.- On line 223, clarify why an analysis of high-density EEG current sources could improve the results obtained.
